# Correlations between social media addiction and anxiety, depression, FoMO, loneliness and self-esteem among students: A systematic review and meta-analysis

Zhang Jing[1], Wang Yang[2], Zhou Lei[1], Wu Junmei[3], Li Hui[4]*, Zhu Tianmin[1]*

1 School of Health Preservation and Rehabilitation, Chengdu University of Traditional Chinese Medicine, Chengdu, China, 2 School of Sports Medicine and Health, Chengdu Sport University, Chengdu, China, 3 School of Acupuncture and Tuina, Chengdu University of Traditional Chinese Medicine, Chengdu, China, 4 School of Preclinical Medicine, Chengdu University, Chengdu, China

* ttlihui@163.com; tianminzhu@cdutcm.edu.cn

## Abstract

### Background and aims

With the ubiquity of the internet, social media have become an essential part of daily life. There are various types of social media, such as Facebook, Twitter, Tik-Tok, WeChat and SNS. Social media addiction (SMA) was found to be significantly associated with mental health concerns, self-esteem, fear of missing out (FoMO), and loneliness on the basis of a literature review concerning SMA. To further explore the connections between SMA and anxiety, depression, self-esteem, FoMO and loneliness, we performed a meta-analysis to quantitatively synthesize the previous findings,

### Methods

The PubMed, Embase, Web of Science, Chinese National Knowledge Infrastructure (CNKI), Chinese Biological Medicine (CBM) and Technology Journal Database (VIP) databases were accessed to perform a systematic review and meta-analysis. This search was updated in April. Pooled Pearson's correlation coefficients between SMA and anxiety, depression, loneliness, FoMO and self-esteem were calculated with STATA software via a random or fixed effects model.

### Results

Thirty-two studies involving a total of 26166 students were identified. The meta-analysis revealed positive correlations between SMA and anxiety, depression, loneliness and FoMO (anxiety: summary r = 0.31, 95% CI = 0.25–0.36, P < 0.001; depression: summary r = 0.31, 95% CI = 0.27–0.34, P < 0.001; loneliness: summary

**Data availability statement:** All relevant data are within the paper and its Supporting Information files.

**Funding:** Funding was provided by the Sichuan Science and Technology Program (Award Number 2024YFFK0160) to Zhu Tianmin, and the Xinglin Scholar Research Promotion Project of Chengdu University of TCM (Award Number XSGG2019007) to Zhu Tianmin.

**Competing interests:** The authors have declared that no competing interests exist.

r = 0.21, 95% CI = 0.13–0.29, P < 0.001; FoMO: summary r = 0.41, 95% CI = 0.36–0.45, P < 0.001). A negative correlation was found between self-esteem and SMA (self-esteem: summary r = -0.24, 95% CI = -0.26– -0.22, P<0.001).

## Conclusions

This meta-analysis revealed that SMA was positively associated with anxiety, depression and loneliness but negatively associated with self-esteem. These findings indicate that students with SMA are more likely to suffer from anxiety, depression and loneliness. Conducting larger prospective studies would be beneficial to verify our findings.

## 1. Introduction

Social media have emerged as a relatively novel concept in the cyber era. With the development of social media technology, space and time have been transcended [1], which can reduce the costs of interpersonal communication. It is a unique technology that allows people to interact with others instantaneously through social media platforms such as Facebook and Instagram [2]. As of January 15, 2025, there were 5.40 billion internet users, equivalent to 66% of the world's population [3]. And the China Internet Network Information Center (CNNIC) published *the 51st Statistical Report on the Development Status of the Internet in China* in March 2022. According to a previous report, there were 1.067 billion internet users in China, an increase of 35.49 million from December 2021, and the internet penetration rate in China was 75.6% [4]. On the one hand, due to the large influx of information, internet users are constantly confirming the validity and trustworthiness of information on social media [5]; on the other hand, this feature increases dependence on social media owing to the increased demand for information [6]. Eventually, continuous social media overuse tends to result in SMA [7,8]. SMA is a behavioral addiction that is characterized by compulsive engagement in social media platforms, resulting in significant disruptions to the user's functioning in crucial life domains, including interpersonal relationships, work or academic performance, and physical health [9,10]. According to previous research [11], social media screen time increased by 51.2% from 2013 to 2021. SMA has in turn become a subject of considerable interest in recent years because of the expansion of digital technology [12].

Substantial prior research has revealed that anxiety and depression are positively related to SMA. Mental health issues may be risk factors for SMA [13,14]. Both longitudinal and cross-sectional studies have shown that anxiety is a major risk factor for internet addiction [15,16]. People with heightened anxiety may become addicted to social media through prolonged, excessive use [17]. In addition, studies have shown that individuals experiencing depressive symptoms often suffer from social media addiction [18–20].

Self-esteem is a subjective evaluation that refers to how people feel about themselves [21] and numerous studies have shown a negative correlation between SMA and self-esteem, which may be related to the greater need for people with low self-esteem to cultivate identity through social media [22]. However, low-self-esteem

individuals perceive more frequent social comparisons on social media platforms and compare themselves to others more frequently, resulting in downward comparisons and self-devaluation [23].

Meanwhile, with the advancement of society and the deepening division of labor, loneliness has emerged as a significant contributing factor to SMA. After extensive research, researchers classified loneliness according to its causes, which can be divided into emotional loneliness and social loneliness [24]. Loneliness is regarded as a state of unmet personal or social emotional needs characterized by feelings of depression, melancholy, low spirits, and emptiness [25], as well as pessimism, separation, and isolation [26,27]. When experiencing social isolation or loneliness, some users develop an emotional attachment to the internet and social media [28].

The information gap between the limited social space in real life and the vast internet space promotes anxiety related to obtaining more information. Consequently, individuals may struggle with the "fear of missing out" (FoMO). FoMO was first defined as "the pervasive fear that others may have positive experiences that they lack" in an early academic study by Przybylski et al. [29].

According to the published research, anxiety, depression, loneliness, FoMO and self-esteem are the most common factors in SMA among adolescents. This article aims to explore the relationship between SMA and anxiety, depression, loneliness, FoMO and self-esteem, in order to understand the etiology of SMA and provide new approaches for prevention or treatment. As a populous country and a developing country that is rapidly digitizing, China is confronted with a severe challenge: the sharp increase in the number of teenagers suffering from SMA has escalated into an urgent public health issue. Therefore, studying the psychological roots of this phenomenon has become an urgent priority. Based on the above purposes, this study can deepen our understanding of addictive behaviors and help prevent adverse effects on students' physical and mental health.

## 2. Materials and methods

This meta-analysis was conducted and reported according to the Preferred Reporting Items for Systematic Review and Meta-Analyses (PRISMA) guidelines [30]. The review protocol was registered with PROSPERO (CRD42023446002).

### 2.1 Search strategy

Studies were found by searching the PubMed, Embase, Web of Science, Chinese National Knowledge Infrastructure (CNKI), Chinese Biological Medicine (CBM) and China Science and Technology Journal (VIP) databases for relevant literature. "Social media*", "SNS", "social networking site*", "TikTok", "Facebook", "Twitter", "Instagram", "WeChat", "Snapchat", "Weibo", "QQ", and "social network site*" were among the search terms used to identify research on social media. The phrases "addiction", "dependen*", "abuse", "disorder", "compulsi*", and "excess" were used to search for conditions. Susceptible populations were identified via the search terms "students", "teenagers", and "adolescents". Anxiety, depression, loneliness, self-esteem, and FoMO have been examined more than other conditions have and were thus chosen for the search after a large amount of literature was reviewed. These search phrases were then merged via the proper Boolean operator. We retrieved all relevant literature with this search method up to April 2024.

### 2.2. Study selection criteria

Two reviewers screened all literature against the following selection criteria to find potentially relevant articles: (a) cross-sectional studies that reported Pearson or Spearman correlation coefficients for associations between SMA and anxiety, depression, loneliness, FoMO, or self-esteem; (b) participants were high school seniors, college students, and graduate students; (c) social media addiction was assessed using robust scales such as the BSMAS, SMDS, BAFS, CSMAS, MTUAS, CSMSM-DS, FIS, SNSAS-8, SMDS, SAS-SV, SMAS, FIQ, SNAQ, SMUQ, and SNI; (d) to assess loneliness, the following instruments were used: UCLA-LS, DJGLS, DLS, LACA, RPLQ, NDLS and SELSA-S; (e) the instruments for assessing self-esteem were restricted to RSES and SISE; (f) the instruments for assessing depression were

restricted to PHQ-9, DASS-21, GAD-7, GHQ-28, SDS, BDI and SDHS; (g) the assessment of anxiety was restricted to the following instruments: PHQ-2, DASS-21, GED, GHQ-28 and SAS; (h) FoMO measures were limited to FoMO-S; (i) conference abstracts and review articles were excluded; (j) literature with low quality or obvious data errors was excluded, i.e., literature with a score of less than 6 on the JBI Critical Appraisal Checklist for Studies Reporting Prevalence Data; and (k) studies with a sample size of less than 200 were excluded.

## 2.3. Data extraction

The data were independently extracted via a form designed specifically for the study. The following information was extracted: first author, year of publication, geographic location, participant's educational level, sample size, cases of male and female participants, mean age, instruments used to measure the degree of SMA, and instruments used to measure levels of anxiety, depression, loneliness, FoMO and self-esteem (see Table 1).

## 2.4. Quality assessment

The methodological quality of all the studies included was independently assessed by two researchers (ZJ and ZL) via the nine-item Joanna Briggs Institution Critical Appraisal Checklist for Studies Reporting Prevalence Data [31] (Appendix A). A minor adjustment was made to the third item. The sample size was determined according to Pearson's correlation rather than prevalence. For ambiguous items, we sought assistance from a third researcher (ZTM) to achieve a consensus. The answers for each item included "yes," "no," "unclear," and "not applicable." An item was assigned a score of one if the answer was "yes." Otherwise, it was assigned a score of zero. Higher scores reflected better methodological quality. Detailed information about the quality assessment is shown in Table 2. All included studies were considered to be of moderate to high quality (total score≥6).

## 2.5. Statistical analysis

Using Pearson product–moment correlation coefficients (r values), the relationships between these factors and SMA were evaluated. We derived the Pearson's and Spearman's correlation coefficients from these investigations. We utilized the following formula to convert Spearman's correlation coefficients into Pearson's correlation coefficients to guarantee the consistency of the findings: $r_s == \frac{6}{\pi} = \sin{-1} = \frac{r}{2}$. To obtain the variance stability of the correlation coefficient, the Pearson correlation coefficients were transformed to a normal distribution via "Fisher's z-transformation" [32]. The $I^2$ statistic was used to investigate statistical heterogeneity. The results showed significant heterogeneity if the $I^2$ value was greater than 50%, and a random effects model was employed to evaluate the data. A fixed effects model was applied otherwise [33]. The meta-analysis was carried out via STATA statistical software, version 17.0.

## 3. Results

### 3.1. Selected studies

After 831 duplicates were deleted, 1516 studies were selected via our search approach. A total of 1315 studies were excluded because of irrelevant research, small sample sizes and conference papers or reviews through screening titles and abstracts. After the full texts of 201 articles were read, 169 articles were removed. The reasons for removal were as follows: 1) insufficient data, 2) duplicate publications, 3) no correct assessment scale, 4) small sample size, 5) no correlation studies, 6) abstract only, and 7) poor quality. For this systematic review, 32 studies were chosen. Fig 1 shows the study selection procedure.

### 3.2 Study design Characteristics

In total, 32 studies [34–65], 9 of which were from China, met the inclusion criteria. All included studies were cross-sectional and included a total of 25719 participants. The design characteristics of the included studies are shown in Table 1.

**Table 1. The characteristics of SMA-related 32 studies included in this meta-analysis.**

| Author | Year | Country | Male/Female | Age(mean±SD) | SMA measurement | Education level(n) | Measurement instrument(Pearson's r) | | | | |
|--------|------|---------|-------------|--------------|-----------------|--------------------|-----------------|-----------|------|------------|---------|
| | | | | | | | Self-esteem | Loneliness | FoMO | Depression | Anxiety |
| Servidio | 2024 | Italy | 66/190 | 23.05±3.58 | BSMAS | University | RSES | N/A | FoMOS | N/A | N/A |
| Akbari | 2023 | Iran | 1112/2263 | 15.46±1.63 | BSMAS | High school | RSES | UCLA-LS | N/A | DASS-21 | DASS-21 |
| Fekih | 2023 | Lebanon | 139/224 | 22.65±3.48 | SMDS | University | N/A | DJGLS | N/A | N/A | N/A |
| Ciacchini | 2023 | Italy | 109/149 | 17.42±1.73 | BSMAS | High school | RSES | N/A | N/A | N/A | N/A |
| Varchetta | 2023 | Span | 118/471 | 21.56±2.73 | BSMAS | University | N/A | N/A | FoMOS | N/A | N/A |
| Al-Mamun | 2022 | Bangladesh | 344/257 | 16.01±5.71 | BAFS | High school(394), Medical college(178), University(29) | N/A | N/A | N/A | GAD-2 | PHQ-2 |
| Chi | 2022 | China | 454/484 | N/A | BSMAS | University | N/A | N/A | FoMOS | N/A | N/A |
| Fabris | 2022 | Italian | 236/236 | 13.50±1.87 | BSMAS | Middle school | N/A | N/A | FoMOS | N/A | N/A |
| Xiao | 2022 | China | 526/496 | 15.12±1.51 | CSMAS | High school | N/A | N/A | N/A | GAD-7 | CED-S |
| Kostic | 2022 | Serbia | 211/346 | 18.09±0.28 | MTUAS | High school | N/A | N/A | FoMOS | N/A/ | N/A |
| Luo | 2022 | China | 282/205 | 18.19±0.829 | CSMSM-DS | University | N/A | UCLA-LS | N/A | N/A | N/A |
| Błachnio | 2021 | Poland | 405/991 | 21.25±4.56 | FIS | High school(494), University(902) | N/A | N/A | N/A | N/A | GHQ-28 |
| Gong | 2021 | China | 447/620 | >18 | SNSAS-8 | University | N/A | UCLA-LS | N/A | PHQ-9 | N/A |
| KılınçelK | 2021 | Turkey | 420/722 | 15.6±2.8 | SMDS | High school | N/A | UCLA-LS | N/A | N/A | STAI |
| Sha | 2021 | China | 1305/1731 | 16.56±0.62 | SAS-SV | High school | N/A | N/A | N/A | DASS-21 | DASS-21 |
| Uyaroğlu | 2021 | Turkey | 84/471 | 30.68±11.45 | SMAS | University | N/A | SELSA-S | N/A | N/A | N/A |
| Wang | 2021 | China | 320/368 | 13.44±0.99 | FIQ | Middle school | RSES | N/A | N/A | N/A | N/A |
| Dadiotis | 2021 | Greece | 59/266 | 21.6±5.26 | BSMAS | University | RSES | UCLA-LS | N/A | DASS-21 | DASS-21 |
| Ahmed | 2021 | Bangladesh | 167/196 | 20.87±1.81 | BSMAS | University | RSES | N/A | N/A | N/A | N/A |
| Acar | 2020 | Turkey | 112/109 | 15.86±0.91 | SMAS | High school | RSES | N/A | N/A | N/A | N/A |
| Fang | 2020 | China | 147/354 | 19.6±1.24 | FIQ | University | N/A | N/A | FoMOS | N/A | N/A |
| Blasco | 2020 | Span | 45/316 | 22.38±10.43 | SNAQ | University(321), Master(33), Doctorate(7) | N/A | N/A | N/A | N/A | BAI |
| Shen | 2020 | China | 173/226 | 20.40±1.35 | SMAQ | University | N/A | N/A | FoMOS | N/A | N/A |
| Bloemen | 2020 | Belgium | 533/298 | 15.94±1.24 | N/A | High school | N/A | N/A | FoMOS | N/A | N/A |
| Pilar | 2019 | Span | 106/172 | 19.4±0.16 | SNAQ | University | RSES | UCLA-LS | N/A | N/A | N/A |
| Yin | 2019 | China | 301/403 | 16.8±0.92 | FIQ | High school | N/A | N/A | FoMOS | N/A | N/A |
| Worsley | 2018 | England | 259/770 | 19.80±1.67 | BSMAS | University | N/A | N/A | N/A | PHQ-9 | N/A |
| Kırcabu-run | 2018 | Turkey | ①418/386 | 16.2±1.03 | SMUQ | High school (804) | RSES | N/A | N/A | SDHS | N/A |
| | | | ②304/456 | 21.48±3.73 | | University (760) | | | | | |
| Oberst | 2017 | Spain | 377/1091 | 16.59±0.62 | SNI | High school | N/A | N/A | FoMO | N/A | N/A |
| Pontes | 2017 | Portugal | 265/244 | 13.02±1.64 | BFAS | Middle school | N/A | N/A | N/A | DASS-21 | DASS-21 |
| Hawi | 2016 | Lebanon | 190/174 | 21.1±2.3 | SMAS | University | RSES | N/A | N/A | N/A | N/A |
| Koc | 2013 | Turkey | 347/100 | 21.64±1.94 | FIQ | University | N/A | N/A | N/A | GHQ-28 | GHQ-28 |

*(Continued)*

**Table 1.** (Continued)

Note: BAFS, The Bergen Facebook Addiction Scale; GAD-2, The two-item Generalized Anxiety Disorder scale; PHQ-2, The two-item Patient Health Questionnaire; BSMAS, The Bergen Social Media Addiction Scale; FoMO, Fear of Missing Out scale; CSMAS,The Chinese Social Media Addiction Scale; GAD-7,Anxiety symptom intensity was measured using the 7-item Generalized Anxiety Disorder Test; CES-D, Depressive symptoms were assessed using the validated Center for Epidemiological Studies Depression Scale; MTUAS, the Media and Technology Usage and Attitudes Scale; CSMSMDS, Mobile social media dependence was measured by College Students' Mobile Social Media Dependence Scale; UCLA-LA, University of California, Los Angeles-Loneliness Scale; SELSA, The Social and Emotional Loneliness Scale for Adults; DJGLS, De Jong Gierveld Loneliness Scale; FIQ, the Facebook Intrusion Questionnaire; GHQ-28, The General Health Questionnaire; SNSAS, The Chinese Social Networking Sites Addiction Scale; PHQ-9,The nine-item Patient Health Questionnaire; SMDS, Social Media Disorder scale; SAS-SV, The Smartphone Addiction Scale, Short Version; DASS-21, The Depression Anxiety Stress Scales 21; RSES, The Rosenberg Self-esteem Scale; SMAS, the Social Media Addiction Scale; SNAQ, the Social Network Addiction questionnaire; BAI, the Beck Anxiety Inventory; SMUQ, Social Media Use Questionnaire; SNI, Social network intensity scale.

**Table 2. Quality assessment of the included 32 studies in the meta-analysis.**

|  | Item1 | Item2 | Item3 | Item4 | Item5 | Item6 | Item7 | Item8 | Item9 | TOTAL |
|---|---|---|---|---|---|---|---|---|---|---|
| Servidio, 2024, Italy | Y | Y | Y | N | Y | Y | Y | Y | Y | 8 |
| Akbari, 2023, Iran | Y | Y | Y | N | Y | Y | Y | Y | Y | 8 |
| Fekih, 2023, Lebanon | Y | Y | Y | N | Y | Y | Y | Y | N | 7 |
| Ciacchini, 2023, Italy | Y | Y | Y | N | Y | Y | N | Y | N | 6 |
| Varchetta, 2023, Span | Y | Y | Y | Y | Y | Y | Y | Y | N | 8 |
| AlMamun, 2020, Bangladesh | Y | Y | Y | Y | Y | Y | Y | Y | N | 8 |
| Chi, 2022, China | Y | Y | Y | N | Y | Y | Y | Y | Y | 8 |
| Fabras, 2022, Italian | Y | Y | Y | N | Y | Y | Y | Y | N | 7 |
| Xiao, 2022, China | Y | Y | Y | N | Y | Y | Y | Y | Y | 8 |
| Kostic, 2022, Serbia | Y | Y | Y | N | Y | Y | Y | Y | N | 7 |
| Luo, 2022, China | Y | Y | N | Y | Y | N | Y | Y | Y | 7 |
| Błachnio, 2021, Poland | Y | Y | Y | N | Y | N | Y | Y | Y | 7 |
| Gong, 2021, China | Y | Y | Y | N | Y | Y | Y | Y | N | 7 |
| Kılınçel, 2021, Turkey | Y | Y | Y | Y | N | Y | Y | Y | N | 7 |
| Sha, 2021, China | Y | Y | Y | N | Y | N | Y | Y | Y | 7 |
| Uyaroğlu, 2021, Turkey | N | Y | Y | Y | Y | Y | Y | Y | N | 7 |
| Wang, 2021, China | Y | Y | Y | N | Y | Y | Y | Y | Y | 8 |
| Dadiotis, 2021, Athens | Y | Y | N | N | Y | Y | Y | Y | N | 7 |
| Ahmed, 2021, Bangladesh | Y | Y | N | N | Y | Y | Y | Y | Y | 7 |
| Acar.i, 2020, Turkey | Y | Y | N | N | Y | Y | Y | Y | N | 6 |
| Fang, 2020, China | Y | Y | Y | N | Y | Y | Y | Y | N | 7 |
| Blasco, 2020, Spanish | Y | Y | N | N | Y | Y | Y | Y | Y | 7 |
| Shen, 2020, China | Y | Y | N | N | Y | Y | Y | Y | N | 6 |
| Bloemen, 2020, Belgium | Y | Y | Y | Y | N | N | Y | Y | Y | 7 |
| Pilar, 2019, Spanish | Y | Y | N | Y | Y | Y | Y | Y | N | 7 |
| Yin, 2019, China | Y | Y | Y | N | Y | Y | Y | Y | N | 7 |
| Worsley, 2018, England | Y | Y | Y | N | Y | Y | N | Y | Y | 7 |
| Kırcaburun, 2018, Turkey | Y | Y | Y | N | Y | Y | Y | Y | N | 7 |
| Oberst, 2017, Spain | Y | Y | N | N | Y | Y | Y | Y | N | 6 |
| Pontes, 2017, Portugal | Y | Y | N | Y | Y | Y | Y | Y | Y | 8 |
| Hawi, 2016, Lebanon | Y | Y | Y | N | Y | Y | Y | Y | N | 7 |
| Koc, 2013, Turkey | Y | Y | N | N | Y | Y | Y | Y | Y | 7 |

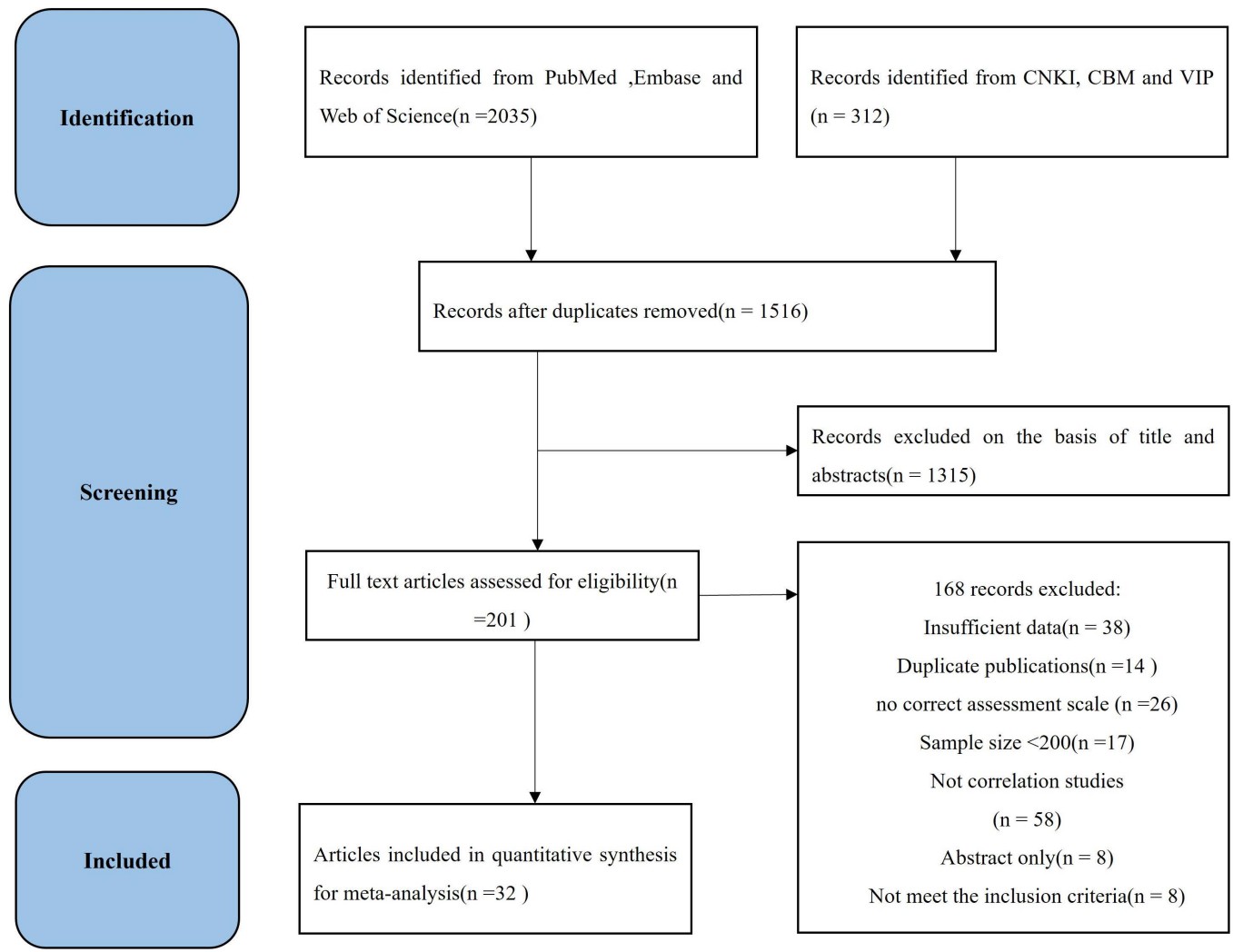

**Fig 1. The flow chart of the study selection process.**

### 3.3. Main outcomes and Meta-Analysis

Heterogeneity tests were conducted for anxiety, depression, loneliness, self-esteem and FoMO. All the results were greater than 50%. Therefore, a random effects model was used for the meta-analysis. The effect value was obtained via the conversion formula and converted into the summary value r. The results are shown in Table 3. The summary r value refers to the pooled effect size.

### 3.4. SMA and anxiety

Nine studies [37–39,43,49,51,56,58,64] reported significant correlation coefficients between SMA and anxiety, encompassing a total of 8,839 participants. According to the random effects model, the pooled effect size (z) was 0.32 (95% CI: 0.25–0.38; see Fig 2). After conversion, the pooled r was 0.31 (95% CI: 0.25–0.36; see Table 3).

**Table 3. The pooled effect size (summary r) after conversion.**

| Factors | Summary r | 95%CI | I² | P |
|---|---|---|---|---|
| Anxiety | 0.31 | (0.25, 0.36) | 87.9% | <0.001 |
| Depression | 0.31 | (0.27, 0.34) | 69.4% | <0.001 |
| Loneliness | 0.21 | (0.13, 0.29) | 90.9% | <0.001 |
| FoMO | 0.41 | (0.36, 0.45) | 79.8% | <0.001 |
| Self-esteem | −0.24 | (−0.26, −0.22) | 80.9% | <0.001 |

### 3.5. SMA and depression

Nine [37,43,47,50,51,56,58,63,64] studies reported significant correlation coefficients between SMA and depression, with a total sample size of 9600. According to the random effects model, the pooled effect size (z) was 0.32 (95% CI: 0.28–0.36; see Fig 3). After conversion, the pooled r was 0.31 (95% CI: 0.27–0.34; see Table 3).

### 3.6. SMA and loneliness

Eight studies [36,43,46,47,49,53,55,60], representing a total sample size of 7592, reported correlation coefficients between SMA and loneliness. We used a random effects model, in which the pooled effect size (z) was 0.21, to carry out our analysis (95% CI: 0.13–0.30; see Fig 4). After conversion, the pooled r was 0.21 (95% CI: 0.13–0.29; see Table 3).

### 3.7. SMA and FoMO

Ten studies [40,41,44,45,52,54,57,59,61,65], representing a total sample size of 6715, reported correlations between SMA and anxiety. A random effects model was used with a combined effect size (z) of 0.44 (95% CI: 0.38–0.49; see Fig 5). After conversion, the combined r is 0.41 (95% CI: 0.36–0.45; see Table 3).

### 3.8. SMA and self-esteem

To conduct a meta-analysis on the relationship between SMA and self-esteem, we identified ten articles that together accounted for a sample size of 7,962 students [34–36,42,43,48,50,55,57,62]. We obtained an outcome pooled effect size (z) of −0.24 (95% CI: −0.27–0.22; see Fig 6) based on a random effects model. After conversion, the pooled r was −0.24 (95% CI: −0.26–0.22; see Table 3).

### 3.9. Publication bias

After one highly heterogeneous study was removed, the funnel plot was basically symmetrical (see Fig 7). Begg's test and Egger's test suggested that there was no publication bias (*P* > 0.05).

### 3.10. Sensitivity analyses

Sensitivity analyses were conducted to assess the stability of our findings via a one-by-one elimination method whereby a single study was removed and the summary correlation coefficients were recalculated.

## 4. Discussion

With the rapid expansion of social media users, SMA has attracted considerable interest in recent years [66,67]. As shown in a recent meta-analysis, the global prevalence of SMA is 24%, and approximately one in five people may be at high risk for SMA [68]. However, the majority of studies have examined only the relationship between a certain aspect and SMA rather than pursuing a more thorough understanding of the relationship between SMA and different psychological or social

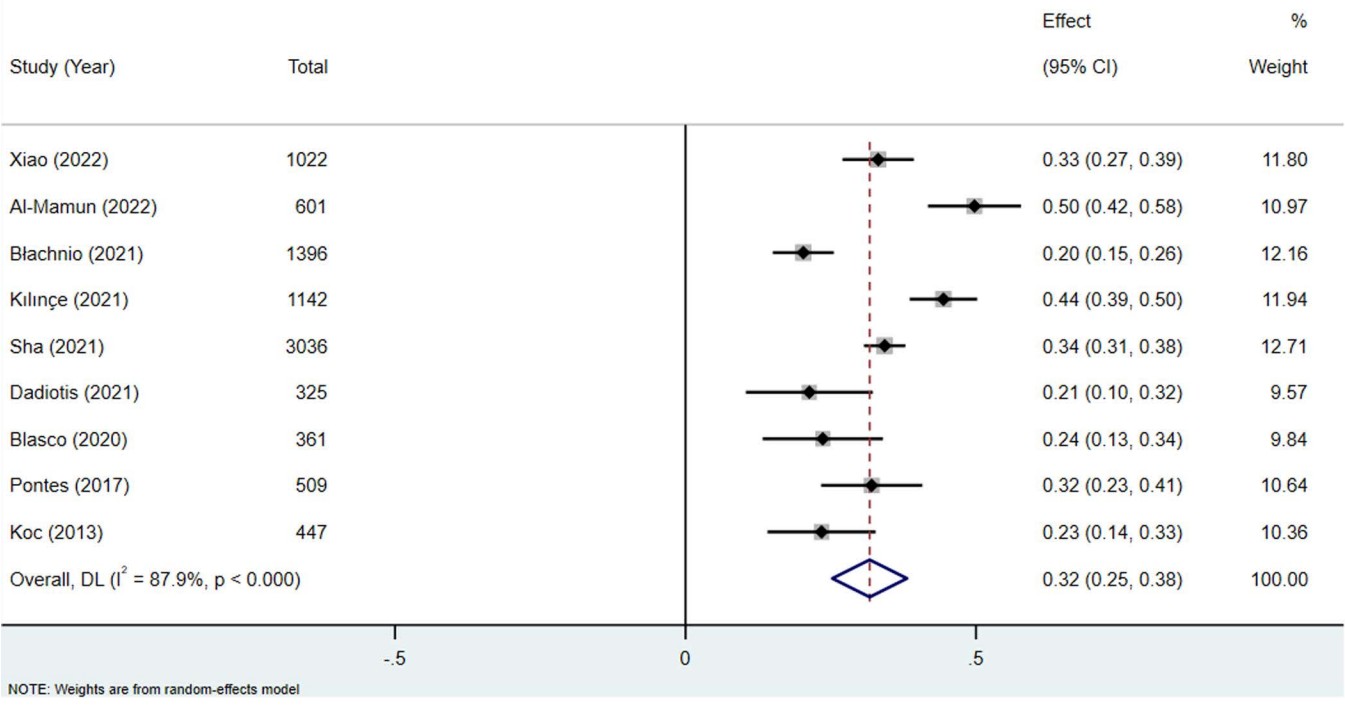

**Fig 2. Meta analysis for the relationship between SMA and anxiety.**

issues. In our group's previous research [69–71], we reported that individuals with internet addiction are more prone to neuropsychological issues such as anxiety, depression, and loneliness. Furthermore, in our investigation of the factors influencing SMA, a substantial body of literature has corroborated the associations between self-esteem and FoMO with SMA. Based on these findings, we selected anxiety, depression, loneliness, self-esteem, and FoMO as variables for a comprehensive analysis investigating the interplay between these psychological factors and SMA. To the best of our knowledge, this was the first meta-analysis exploring the summary correlation coefficients of SMA with anxiety, depression, loneliness, FoMO and self-esteem. Our results revealed weak to intermediate positive correlations between SMA and anxiety, depression, loneliness and FoMO, with summary Pearson's correlation coefficients of 0.31, 0.31, 0.21 and 0.41, respectively. Additionally, self-esteem showed a weak negative correlation with SMA, with a summary Pearson's correlation coefficient of −0.24. All the 95% CIs in the sensitivity analyses ranged from 0–1, which indicated that the correlation coefficients were reliable and convincing. The current meta-analysis offers strong evidence that low self-esteem, anxiety, depression, loneliness, and the fear of missing out are all positively correlated with SMA. Thus, students with SMA are more likely to display features of severe anxiety, depression, loneliness, FoMO and low self-esteem. SMA has been linked to poor academic performance [72] and job burnout [73], which has a negative impact on students' academic and career development. The internet is a double-edged sword, and more studies are needed to determine how to use internet resources properly to reduce the negative impact on students' career development and physical and mental health. In the future, we may explore the behavioral traits of SMA among students in this study from the standpoint of relevant factors to better understand behavioral addiction. Furthermore, this article can provide more comprehensive guidance for various approaches to intervention, government policy-making and the classification of addictive behavior. Enhance the public's

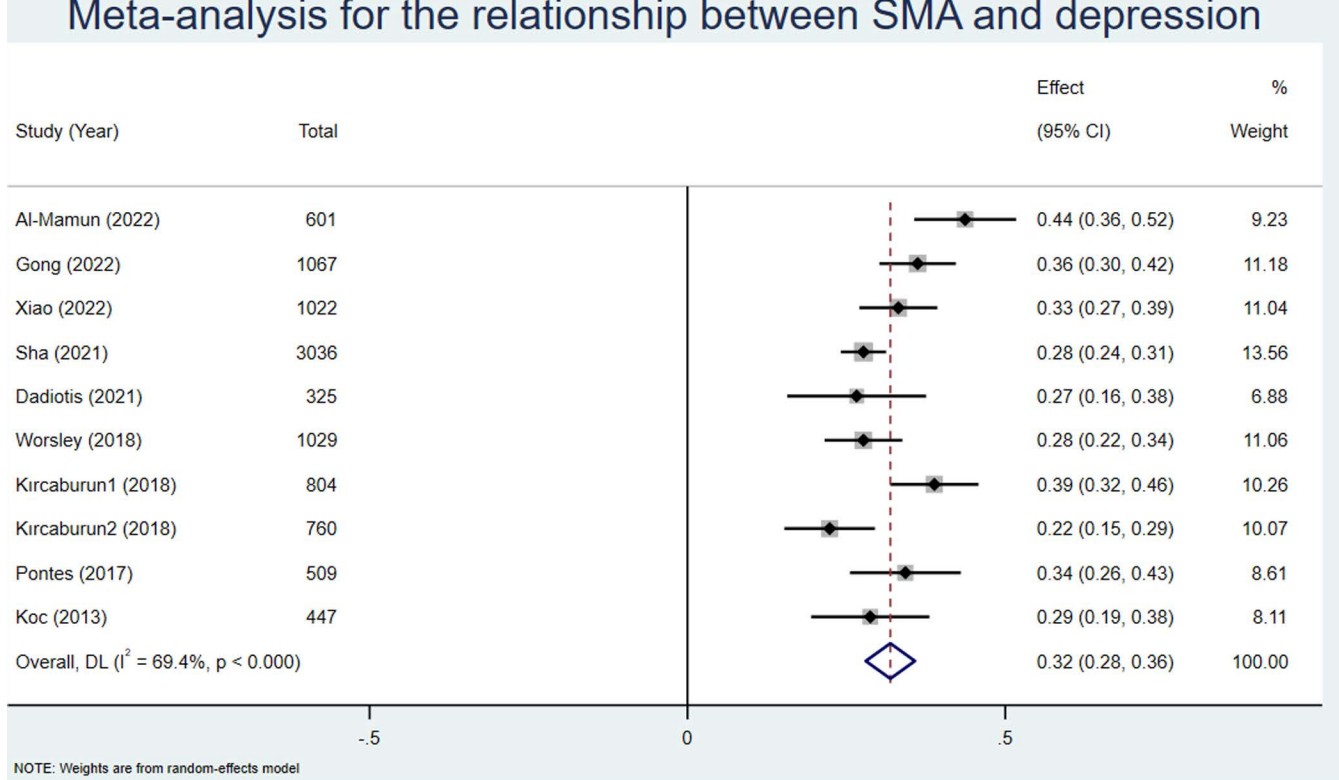

**Fig 3. Meta analysis for the relationship between SMA and depression.**

awareness of preventing Internet addiction, improve their online media literacy and protection skills, and ensure healthy and civilized Internet use.

### 4.1. SMA and anxiety and depression

In our meta-analysis, we analysed the correlation between anxiety and SMA. We collected 11 articles, which included a total sample size of 8,839 students from seven countries: China, Poland, Bangladesh, Turkey, Greece, Spain, and Portugal. The correlation between depression and SMA was investigated by nine articles covering six countries, including China, Bangladesh, Turkey, Greece, Portugal and England, and a total sample size of 9600. Research has shown that nearly half of all cases of depression and anxiety occur in the same patients at the same time [74]. In line with this finding, the relationships between anxiety or depression and SMA were analyzed together. Our meta-analysis revealed that anxiety and depression have similar summary correlation coefficients with SMA. There are two prominent causal explanations for this correlation. Psychopathology (depression, anxiety) can cause SMA because seeking consolation through excessive social media use is a typical sign of depression and anxiety [75]. Moreover, the accessibility of social media applications has increased with advancements in smartphone technology, making social media an essential part of daily life [76]. Relatedly, some studies have shown that teenagers with preexisting mental health issues may use social media to relieve themselves of stressful symptoms via online connections [77]. This behavior is one of the most significant causes of anxiety and depression and is positively correlated with SMA. Sleep quality plays a role in mediating SMA, anxiety and depression [78]. Sleep quality is a key factor in the biological mechanism of emotion regulation [79]. Poor sleep quality is

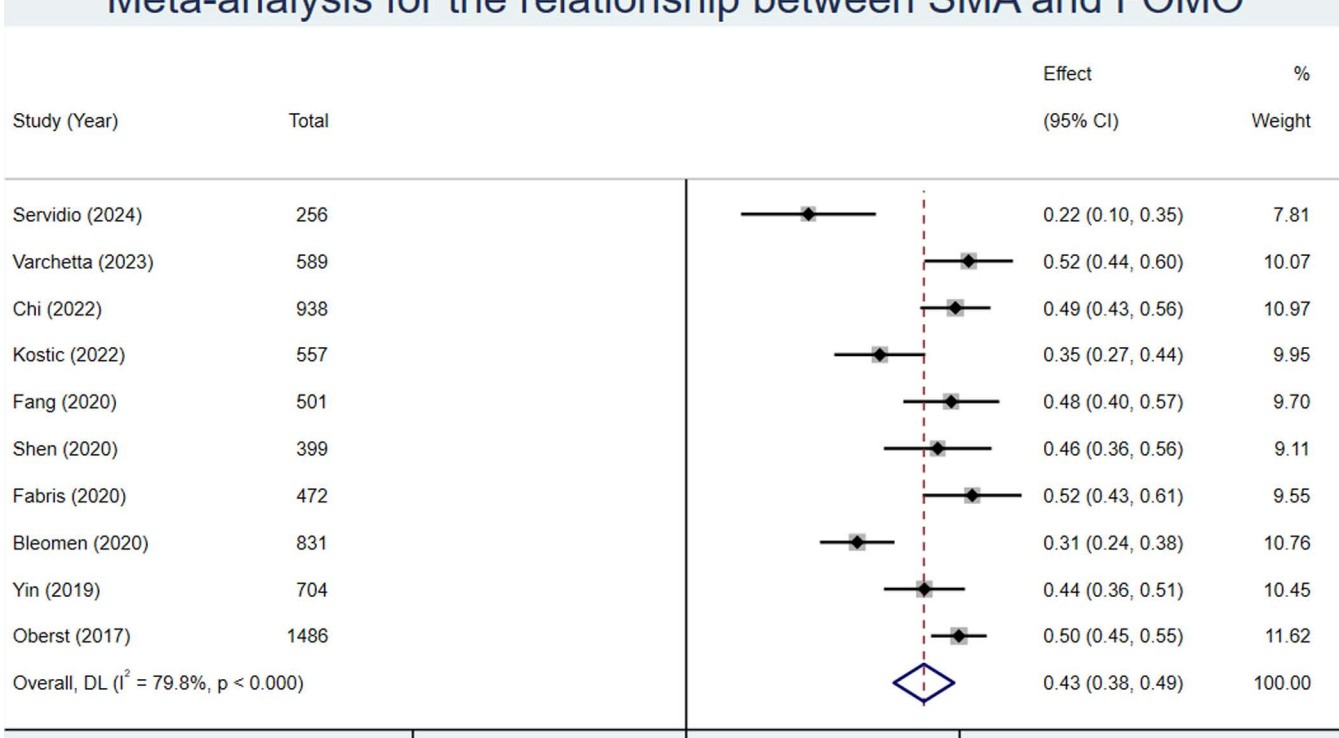

**Fig 4. Meta analysis for the relationship between SMA and anxiety.**

more likely to result in a psychopathological state [80]. People who are addicted to smartphones tend to postpone bedtime, which contributes to increased depression and anxiety [81]. In summary, this bidirectional relationship may eventually generate a vicious cycle between SMA and psychopathology (depression, anxiety).

## 4.2. SMA and loneliness

The analysis included six countries, China, Lebanon, Iran, Turkey, Spain and Greece, and the study population comprised students. The results revealed a positive correlation between loneliness and SMA, in line with the findings of Rebisz's study, which reported a statistically significant bilateral positive correlation: the higher the level of internet addiction was, the stronger the feeling of loneliness was, and vice versa [82]. Loneliness is a painful emotion that can directly or indirectly lead to SMA. Loneliness can be directly alleviated through seeking comfort through social media; it can be indirectly alleviated trough sleep deprivation, which serves as a mediator between loneliness and SMA [83,84]. Sleep deprivation increases the amount of time spent on social media platforms, whereas overreliance on social media causes sleep deprivation and makes people feel lonely. Thus, loneliness and SMA have a positive bidirectional relationship, and the absence of interventions may cause the condition to continue to worsen.

## 4.3. SMA and FoMO

We retrieved 10 studies, including 4 Chinese studies and 6 studies from Spain, Serbia, Italy and Belgium, to examine the correlation between FoMO and SMA. We carried out a meta-analysis that revealed a moderate correlation between FoMO

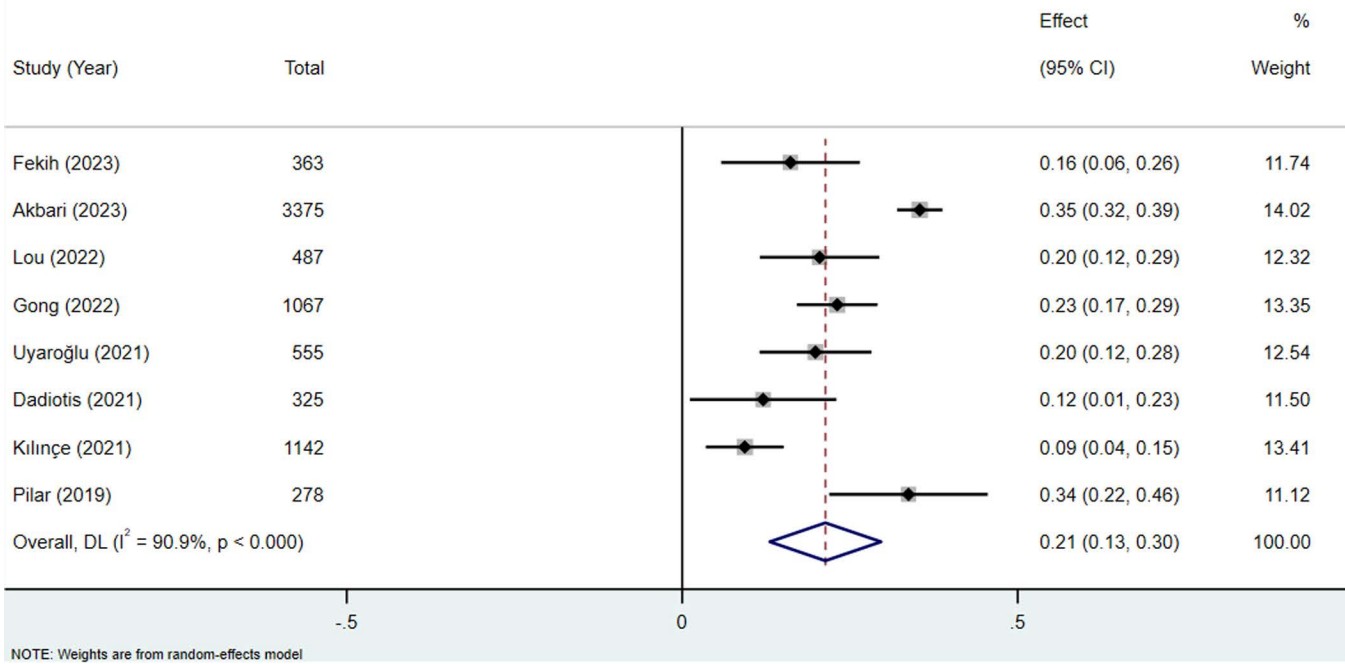

## Meta-analysis for the relationship between SMA and loneliness

| Study (Year) | Total | Effect (95% CI) | % Weight |
|---|---|---|---|
| Fekih (2023) | 363 | 0.16 (0.06, 0.26) | 11.74 |
| Akbari (2023) | 3375 | 0.35 (0.32, 0.39) | 14.02 |
| Lou (2022) | 487 | 0.20 (0.12, 0.29) | 12.32 |
| Gong (2022) | 1067 | 0.23 (0.17, 0.29) | 13.35 |
| Uyaroğlu (2021) | 555 | 0.20 (0.12, 0.28) | 12.54 |
| Dadiotis (2021) | 325 | 0.12 (0.01, 0.23) | 11.50 |
| Kılınçe (2021) | 1142 | 0.09 (0.04, 0.15) | 13.41 |
| Pilar (2019) | 278 | 0.34 (0.22, 0.46) | 11.12 |
| Overall, DL ($I^2$ = 90.9%, p < 0.000) | | 0.21 (0.13, 0.30) | 100.00 |

NOTE: Weights are from random-effects model

**Fig 5. Meta analysis for the relationship between SMA and FoMO.**

and SMA. FoMO has been defined as "a pervasive apprehension that others might be having rewarding experiences from which one is absent" [29]. By definition, FoMO has a direct influence on the usage rate of social media, just as Bakioğlu [85] indicated that FoMO has a direct effect on SMAs. Empirical studies have also revealed that individuals with higher FoMO are more vulnerable to social media abuse [86]. Another possible explanation is that positive meta-cognition plays an intermediary role between FoMO and SMA [87]. A high FoMO score accelerates the formation of addictions, leading to physical and mental fatigue as well as social burnout.

### 4.4. SMA and self-esteem

Ten articles covering eight countries were included in this meta-analysis, with a total sample of 7692 students. The RSES scale is used in the literature to evaluate self-esteem. The results showed that SMA was negatively correlated with self-esteem, indicating a positive correlation with low self-esteem. Two possible explanations regarding the consequences of these correlations are demonstrated below. People with low self-esteem or a poor self-image may prefer to communicate online instead of face-to-face. Research has shown that people with low self-esteem are more apt to believe that social media can make it safer to express themselves than are people with high self-esteem [88]. In line with the research of Mehdizade, the results showed that people with lower self-esteem were more active on social networks and had more self-promoting content in their social network profiles [89]. Another study revealed that teenage addiction to mobile social media is highly predictable by peer pressure and that this association is especially strong in young people who have low self-esteem [90]. However, directionality is impossible to discern because of the cross-sectional nature of the data. SMA may therefore be a consequence or a predictor of low self-esteem.

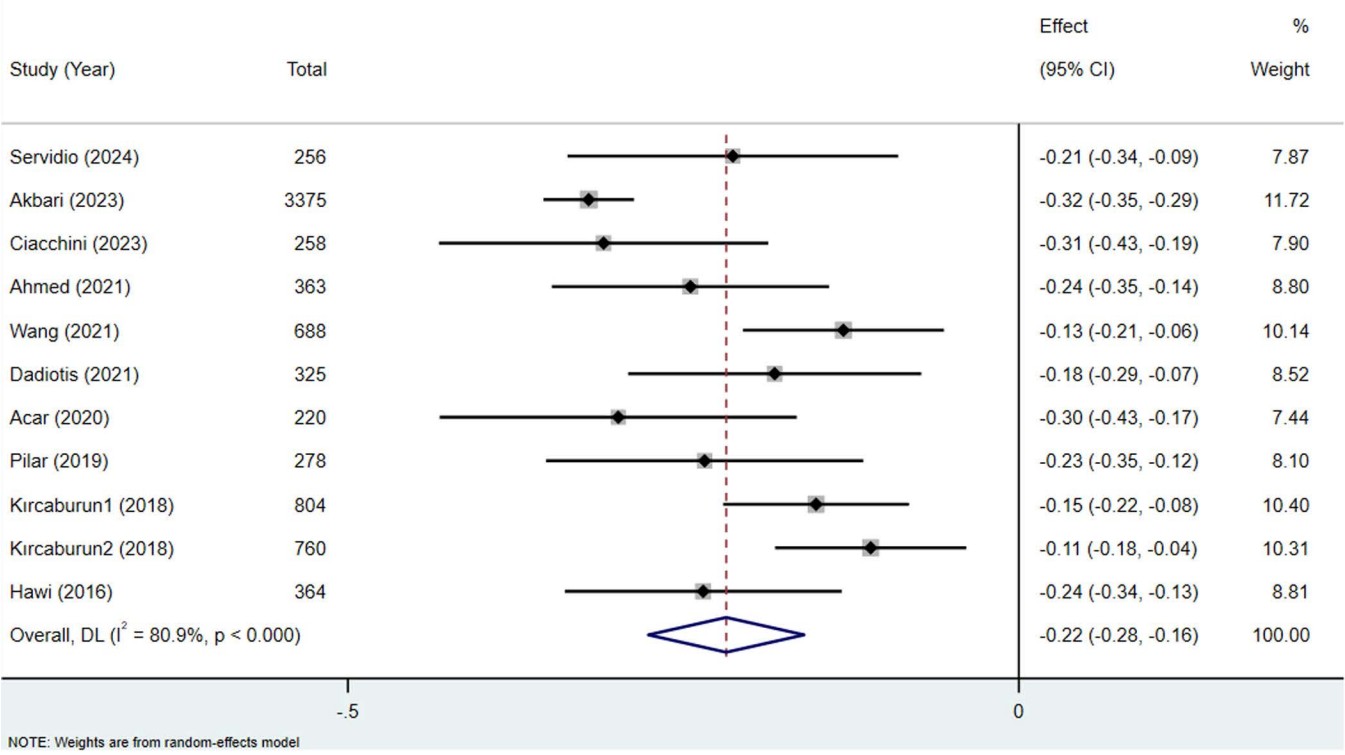

**Fig 6. Meta analysis for the relationship between SMA and self-esteem.**

## 4.5 Strength and limitations

All the studies examined in this meta-analysis were rated as moderate to high quality. The included articles were from diverse countries, which makes it easier to comprehend partial national trends in relevant factors regarding SMA. In the future, data from more nations should be collected to understand global trends in this area of study. Nevertheless, some limitations of the current meta-analysis should be acknowledged. There was significant heterogeneity in the estimation of the relationships between anxiety, depression, loneliness, FoMO, self-esteem and SMA. In the quality assessment, a positive response to Item 4 (Were the study subjects and the setting described in detail?) was found for only 8 out of the 32 included studies. The lack of information about the participants and the environment may hinder us from analyzing the deeper reasons for the research results. Third, we included only cross-sectional research with large sample sizes and excluded studies with small sample sizes. This selection may have had an impact on the comprehensiveness of the analysis. Finally, we are unable to draw conclusions about the direction of causality because the meta-analysis was based on cross-sectional studies and did not consider longitudinal studies. The correlations found might be due to reverse causality. Sometimes, casual relationships may be bidirectional. Future research should include more longitudinal studies that investigate the causal connection between various variables and SMA.

## 4.6. Research significance

In terms of theory, the results of the meta-analysis can provide a more comprehensive understanding of the relevant factors of SMA, filling a gap in research on addictive behaviors. Moreover, the links between SMA and anxiety, depression,

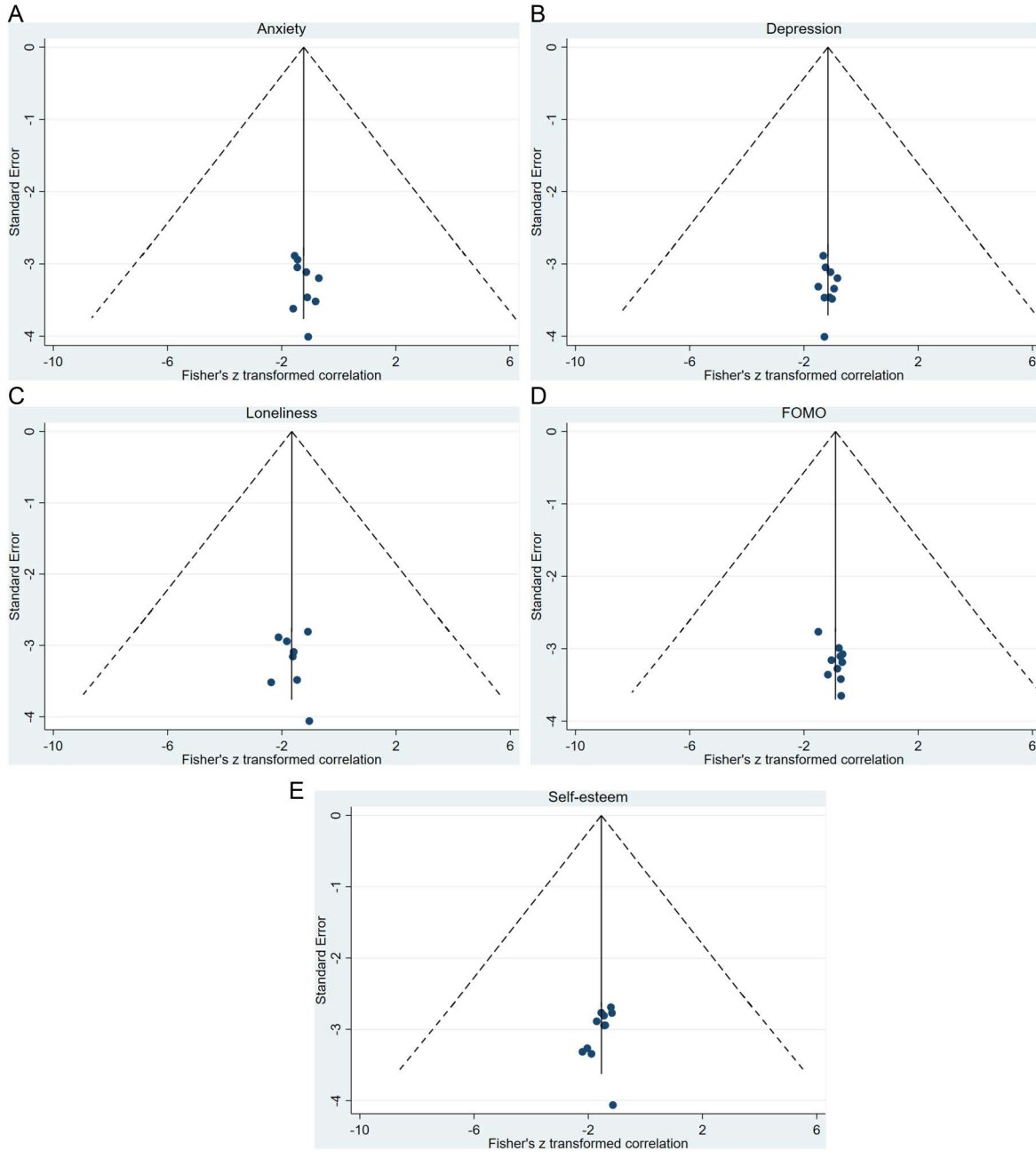

**Fig 7. Funnel plots for the relationship between and (A) anxiety, (B) depression, (C) loneliness, (D) FoMO, and (E) self-esteem.**

loneliness, FoMO, and self-esteem have the potential to expand current theories. From a practical standpoint, the findings of this study provide new insights and recommendations for future intervention approaches, medical treatments, and policy-making with respect to addictive behavior. Furthermore, governments and organizations should be urged to strengthen the self-regulation of online platforms such as social media to protect users from potentially harmful material and lessen the negative impact of social media on students.

## 5. Conclusion

This meta-analysis found a positive link between SMA and depression, anxiety, FoMO, loneliness, and low self-esteem symptoms. Compared with the other four dimensions, the fear of missing out dimension had a greater association with SMA. The cornerstone of individuals' development is healthy psychological growth, and adolescents and students are high-risk populations for SMA. Schools, families and society should help students use social media properly, given that this period is crucial to their development. However, methodological constraints include (a) underpowered sample cohorts in the meta-analysis, and (b) ethnocentric recruitment practices in the source studies, failing to represent cross-cultural populations. We will do our best to make improvements in the future work.

## Supporting information

**S1 File. Cover page.**
(DOCX)

**S2 File. Editing Certificate.**
(PDF)

**S3 File. Information extraction form.**
(DOCX)

**S4 File. Minimum data set.**
(DOCX)

**S5 File. PRISMA 2020 checklist.**
(DOCX)

**S6 Appendix. JBI critical appraisal checklist for studies reporting prevalence data.**
(DOCX)

## Author contributions

**Conceptualization:** Zhang Jing.

**Data curation:** Zhou Lei, Wu Junmei.

**Funding acquisition:** Li Hui, Tianmin Zhu.

**Investigation:** Wang Yang.

**Methodology:** Zhang Jing.

**Software:** Zhou Lei.

**Supervision:** Wang Yang, Li Hui, Tianmin Zhu.

**Validation:** Wu Junmei.

**Writing – original draft:** Zhang Jing.

**Writing – review & editing:** Li Hui, Tianmin Zhu.

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
