## [Decision Letter · Decision Letter 0]

10 Sep 2024

Dear Dr. Zhu,

Thank you for submitting your manuscript to PLOS ONE. After careful consideration, we feel that it has merit but does not fully meet PLOS ONE’s publication criteria as it currently stands. Therefore, we invite you to submit a revised version of the manuscript that addresses the points raised during the review process.

We look forward to receiving your revised manuscript.

Kind regards,

Hua Pang

Academic Editor

PLOS ONE

Journal Requirements: When submitting your revision, we need you to address these additional requirements. 1. Please ensure that your manuscript meets PLOS ONE's style requirements, including those for file naming. The PLOS ONE style templates can be found at https://journals.plos.org/plosone/s/file?id=wjVg/PLOSOne_formatting_sample_main_body.pdf and https://journals.plos.org/plosone/s/file?id=ba62/PLOSOne_formatting_sample_title_authors_affiliations.pdf 2. As required by our policy on Data Availability, please ensure your manuscript or supplementary information includes the following:  A numbered table of all studies identified in the literature search, including those that were excluded from the analyses.   For every excluded study, the table should list the reason(s) for exclusion.   If any of the included studies are unpublished, include a link (URL) to the primary source or detailed information about how the content can be accessed.  A table of all data extracted from the primary research sources for the systematic review and/or meta-analysis. The table must include the following information for each study:  Name of data extractors and date of data extraction  Confirmation that the study was eligible to be included in the review.   All data extracted from each study for the reported systematic review and/or meta-analysis that would be needed to replicate your analyses.  If data or supporting information were obtained from another source (e.g. correspondence with the author of the original research article), please provide the source of data and dates on which the data/information were obtained by your research group.  If applicable for your analysis, a table showing the completed risk of bias and quality/certainty assessments for each study or outcome.  Please ensure this is provided for each domain or parameter assessed. For example, if you used the Cochrane risk-of-bias tool for randomized trials, provide answers to each of the signalling questions for each study. If you used GRADE to assess certainty of evidence, provide judgements about each of the quality of evidence factor. This should be provided for each outcome.   An explanation of how missing data were handled.  This information can be included in the main text, supplementary information, or relevant data repository. Please note that providing these underlying data is a requirement for publication in this journal, and if these data are not provided your manuscript might be rejected.  3. We note that the grant information you provided in the ‘Funding Information’ and ‘Financial Disclosure’ sections do not match.  When you resubmit, please ensure that you provide the correct grant numbers for the awards you received for your study in the ‘Funding Information’ section. 4. We note that your Data Availability Statement is currently as follows: All relevant data are within the manuscript and its Supporting Information files. Please confirm at this time whether or not your submission contains all raw data required to replicate the results of your study. Authors must share the “minimal data set” for their submission. PLOS defines the minimal data set to consist of the data required to replicate all study findings reported in the article, as well as related metadata and methods (https://journals.plos.org/plosone/s/data-availability#loc-minimal-data-set-definition). For example, authors should submit the following data: - The values behind the means, standard deviations and other measures reported;- The values used to build graphs;- The points extracted from images for analysis. Authors do not need to submit their entire data set if only a portion of the data was used in the reported study. If your submission does not contain these data, please either upload them as Supporting Information files or deposit them to a stable, public repository and provide us with the relevant URLs, DOIs, or accession numbers. For a list of recommended repositories, please see https://journals.plos.org/plosone/s/recommended-repositories. If there are ethical or legal restrictions on sharing a de-identified data set, please explain them in detail (e.g., data contain potentially sensitive information, data are owned by a third-party organization, etc.) and who has imposed them (e.g., an ethics committee). Please also provide contact information for a data access committee, ethics committee, or other institutional body to which data requests may be sent. If data are owned by a third party, please indicate how others may request data access. 5. PLOS requires an ORCID iD for the corresponding author in Editorial Manager on papers submitted after December 6th, 2016. Please ensure that you have an ORCID iD and that it is validated in Editorial Manager. To do this, go to ‘Update my Information’ (in the upper left-hand corner of the main menu), and click on the Fetch/Validate link next to the ORCID field. This will take you to the ORCID site and allow you to create a new iD or authenticate a pre-existing iD in Editorial Manager.

Reviewers' comments:

Reviewer's Responses to Questions

**Comments to the Author**

1. Is the manuscript technically sound, and do the data support the conclusions?

Reviewer #1: Yes

Reviewer #2: Partly

2. Has the statistical analysis been performed appropriately and rigorously?

Reviewer #1: Yes

Reviewer #2: Yes

3. Have the authors made all data underlying the findings in their manuscript fully available?

Reviewer #1: Yes

Reviewer #2: Yes

4. Is the manuscript presented in an intelligible fashion and written in standard English?

Reviewer #1: Yes

Reviewer #2: No

Reviewer #1: Comments to the Author(s):

Reviewer Comments on Manuscript ID: PONE-24-20428 (research article)

Thank you for the opportunity to review this manuscript. This study investigated the correlations between social media addiction and various psychological factors such as anxiety, depression, fear of missing out (FOMO), loneliness, and self-esteem through a meta-analysis. Below are suggestions to further improve this interesting and well-written study.

Overall:

1. This is an interesting study and a well-written manuscript.

2. A few grammatical errors were identified that can be addressed through language editing.

3. Adding more literature to the rationale (introduction) and discussion sections will strengthen the manuscript.

The title:

1. The title is clearly formulated and unambiguous, effectively conveying the focus of the study.

The abstract:

1. The abstract clearly outlines the research problem, research methodology, research processes followed in the study, relevant findings, and the implications of the study.

2. There are no page numbers in this manuscript, which may make it difficult to highlight areas for potential revision. Please consider page 1 as where the abstract is located, with page 22 marking the beginning of the reference list. Suggestions for the abstract are:

a. Page 1, line 16: Avoid using the phrase “and so on” as it lacks academic tone.

b. Page 1, line 16: Please clarify what “It” refers to. Does “it” refer to the Internet, social media usage or social media addiction?

c. Page 1, line 16: Replace “tight links” with more academic language such as “significant associations”.

d. Page 1, line 16: Please remove the “the” before psychological issues.

e. Page 1, line 17: Consider using “mental health concerns” instead of “psychological issues”.

f. Page 1, line 18: First mention social media addiction, then follow with the abbreviation (SMA).

g. Page 1, line 19: Please clarify what “these factors” refer to.

h. Page 1: Consider explaining why the combination of anxiety, depression, loneliness, self-esteem, and FOMO was investigated. This should also be clarified in the rationale and discussion sections.

i. Page 2, lines 9 – 13: Please rephrase these sentences for clarity, as the arguments currently do not make sense. Language editing may help here.

Introduction:

1. The introduction/rationale section was approached with interest. Thank you.

2. The rationale section presents valid arguments and the importance of the study is clearly indicated and well-argued.

3. Please ensure that important arguments are supported by citations. Please see:

a. Page 2, line 22.

b. Page 2, line 28.

c. Page 2, line 29.

d. Page 3, line 7.

e. Page 4, line 4.

f. Page 17, line 27.

g. Page 18, line 6.

h. Page 18, line 7.

4. Page 3: An argument is made that the number of social media addicts is increasing. What studies or statistics support this argument? The authors have not provided any statistics or references to substantiate this important argument.

5. Page 3, line 7: Please avoid the phrase “hot topic”, as it lacks academic tone. Instead, refer to it as “a subject of significant interest”.

6. Page 3, line 12: The phrase “are fewer opportunities” should perhaps be “there were fewer opportunities,” as the lockdown is now over

7. Is social media addiction a global concern regardless of COVID-19? The rationale has a strong focus on the pandemic period, but it would be beneficial to highlight whether SMA is a global concern outside of the COVID-19 context. It is unclear whether the authors intend to focus solely on the COVID-19 period.

8. Page 4, lines 17 to 18: Please rephrase for better clarity.

9. Page 4, line 22: Why is the emphasis only on developing effective interventions for teenagers? Many studies indicate that university/college students are the highest users of social media. Why not also advocate for effective interventions for these students?

10. Page 3, line 20: Consider using “mental health issues” rather than “mental issues”.

11. The rationale for the combination of anxiety, depression, loneliness, self-esteem, and FOMO remains unclear after reading this section.

12. Strengthen the rationale section by incorporating more literature to support the arguments presented.

Methodology:

1. The methodology section was discussed properly.

2. Page 5, line 10: Clarify what the six elements refer to. Please elaborate.

3. Page 5, line 24 and line 27: Reference is made to self-esteem in line 24 and then again in line 27. Should self-esteem be replaced with FOMO in line 27?

4. The manuscript indicates that 32 studies were included in the meta-analysis. However, only 31 studies are listed in Table 1.

5. Discrepancies exist between Tables 1 and 2. In Table 1, a study is listed as Fabris (2022) – Italian, which does not appear in Table 2. In Table 2, two studies are listed as Kitiş (2022) – Turkey, and Koc (2013) – Turkey, which do not appear in Table 1. Please ensure that the correct studies are reported on.

Results:

1. The presentation of the results was done in a systematic and structured manner. The results were presented using figures and tables and were discussed properly.

2. Page 12, line 18: Rather refer to the fact that 1315 studies were excluded rather than “should not have been included”.

3. Page 13, line 1: Rather refer to “screening” the titles and abstracts of the sources rather than “browsing”.

4. Page 13, line 2: What does “researches” mean”

5. Page 13, line 5: Reference is made to 27 studies whereas 32 studies are referred to in the manuscript.

6. Page 13, line 13: … nine studies that reported….

7. The rationale for combining anxiety, depression, loneliness, self-esteem, and FOMO remains unclear in this section as well.

8. Page 16, line 13: Is the abbreviation SMA missing from the title of the figure?

Discussion:

1. The discussion of the findings was done well.

2. Page 16, line 22: Refer back to the earlier comment regarding the use of the term “hot topic”.

3. Page 17, line 1: Use “results” instead of “result”.

4. Page 17, line 1: Clarify what the authors consider to be weak correlations. For instance, a correlation of .41 could be considered moderate.

5. Page 17, line 1: What does MPA refer to?

6. Page 17, lines 9 and 10: Should FOMO not form part of this argument?

7. Page 17, line 20: Provide brief information on the countries where the studies were conducted, as well as the sample sizes, similar to how it was done in other sections.

8. Page 19, line 4: Please rephrase.

9. Page 19, line 17: Do you mean the RSES?

10. Page 21, line 2: Consider using the term depression throughout instead of terms such as sadness or despair.

11. Including more literature to support the arguments presented would enhance the overall quality of the manuscript.

Reviewer #2: Thank you, author/s, for coming up with such interesting topic. In order to further improve the manuscript please try to address the following points.

General comments

1. I strongly suggest the author/s to follow the journal guideline. Please start each main heading on new page, use continuous line number, do not start on each page, use Fig instead of Figure.

2. Do not use abbreviation in Title and Abstracts. If abbreviations are used in the main text, it should be used consistently and expanded at first appearance.

3. In order to help readers meaningfully understand their work, authors are expected to present their result in a more clear and intelligible fashion. Look at these sentences taken from this document

‘’Instruments measuring loneliness were limited to…’’ ‘’Instruments measuring self-esteem were limited to...’’ ‘’Instruments measuring depression were limited to..’’ ‘’Instruments measuring anxiety were limited to.

‘’Nine studies mentioned the meaningful correlation coefficients between SMA and anxiety, with the number of the samples was 8839’’

‘’Nine studies mentioned the meaningful correlation coefficients between SMA and depression, with the number of the samples was 9600’’.

‘’Eight studies mentioned the correlation coefficients between SMA and loneliness, with the meaningful number of the samples was 7592’’ So, author/s are suggested to revise the document in this regard.

Abstract

Methods: is both random effect model or fixed effect model used in this study?

Conclusion: I think conclusion should be drawn from the result/finding. However, the conclusion is overstated in both abstract and main text. For e.g. look at these sentences ‘’This meta-analysis can offer prevention for the social media overuse through observing the correlation between these factors and SMA. The viewpoints also can give the direction for correct interventions for the social media addiction’’. As currently written, it seems that the authors have evaluated some intervention methods and suggesting the best among them, which is out of the scope of the study.

Methods

2.2 Study selection criteria: the sentence ‘’Social media addiction was assessed by accepted scale’’ what do you mean by accepted scale? I suggest the author to use more appropriate words

please revise: (h)Self-esteem measures were limited to the FOMO-S

Results

There was contradicting information on the number of articles included in the final analysis, 27 or 32? Please go through your document and revise it. the number of records excluded after full text assessed for eligibility =169 in narration and 168 in diagrammatic flow chart, please revise the inconsistency

I didn’t see any subgroup analysis carried out by authors despite there is significant heterogeneity between studies. to identify the source of this heterogeneity I think the author/s have to conduct sub group analysis for e.g. based on country, measuring scale, educational status etc.

3.8 SMA and self-esteem: please revise this sentence ‘’To start the meta-analysis about the relationship between SMA and self-esteem, were ten studies that have 7962 students in total’’.

Discussion

4.3 SMA and FoMO: line number 3 “The result was significantly higher than other factors’’ instead you can consider to discuss in terms of the strength of correlation without comparing it to other factor and saying lower or higher.

In line number 4 and elsewhere in discussion please try to avoid unnecessary detail for e.g such as the following ‘’The potential explanation for this correlation were discussed further below’’

In discussion section please try to avoid the term “ firsly, secondly’’ hence, it seems you are giving order/rank

4.4 SMA and self-esteem

‘’Studies from ten countries were included in this meta-analysis with a research

16 sample of 7692 students from seven literatures.’’ 1. Please replace literature with articles 2. How seven articles from ten countries?

Strength and limitation

please delete ‘’one strength of this MA’’

Line 5 please delete ‘’especially in terms of statistical analysis of the data and a large enough sample size’’

‘’Additionally, the number of countries engaged (including Turkey, Spain, Portugal, Lebanon, Athens, China, Belgium, Spain, England, Serbia, Poland, Iran, Lebanon and Bangladesh) makes it easier to comprehend partial national trends in relevant factors regarding SMA’’. Please modify it as follows: The included articles were from diverse countries which makes it easier to comprehend partial national trends in relevant factors regarding SMA

Line 11-14 ‘’there was significant heterogeneity in the estimation of the relationships between anxiety, depression, loneliness, FoMO, self-esteem and SMA. This heterogeneity may be due to differences between research methods (including research design and research methods)’’ 1. I think it is not appropriate place to discuss the source of heterogeneity 2. How research design could be the source of heterogeneity when all included articles were cross sectional?

**Do you want your identity to be public for this peer review?** For information about this choice, including consent withdrawal, please see our Privacy Policy

Reviewer #1: **Yes: ** Jacques Jordaan

Reviewer #2: No

---

## [Author Response · Author response to Decision Letter 1]

5 Nov 2024

Dear Editor,

I hope this correspondence finds you in good health and high spirits.

We have supplemented our literature search by identifying all studies, including those excluded from the analysis. For each excluded study, we have listed the reasons for exclusion.

Additionally, we have included tables and the minimum dataset extracted from the primary study sources used for the systematic review and/or meta-analysis.

We have resubmitted the financial disclosure forms, ensuring that the funding information provided in the "Funding" and "Financial Disclosure" sections matches

Thank you for your time and consideration. I look forward to your feedback and the opportunity to contribute to the esteemed journal.

Dear Reviewer 1,

Thank you for your careful consideration of our manuscript “Correlations between social media addiction and anxiety, depression, FoMO, loneliness and self-esteem among students: A systematic review and meta-analysis”. We are grateful for the constructive feedback provided by the reviewers, which we believe has significantly improved the quality and clarity of our work. The black font is your suggestion, while the blue font is the change we made based on your suggestion

The title:

1. The title is clearly formulated and unambiguous, effectively conveying the focus of the study.

The abstract:

1. The abstract clearly outlines the research problem, research methodology, research processes followed in the study, relevant findings, and the implications of the study.

2. There are no page numbers in this manuscript, which may make it difficult to highlight areas for potential revision. Please consider page 1 as where the abstract is located, with page 22 marking the beginning of the reference list. Suggestions for the abstract are:

Reply: We appreciate the reviewers' detailed and helpful suggestions on the abstract. In the lower left corner of the article, we marked the page number. According to your comments, we have made modifications which are displayed in Table 1. Thank you again for your positive comments and valuable suggestions to improve the quality of our manuscript.

Table 1: A description of modifications to the abstract section

a. Page 1, line 16: Avoid using the phrase “and so on” as it lacks academic tone. Page 1, Line 26: We have delete “and so on” and supplemented with specific social networking platforms (SNS).

b. Page 1, line 16: Please clarify what “It” refers to. Does “it” refer to the Internet, social media usage or social media addiction?

Page 1, Line 26: It refer to the social media addiction. We have used the social media addiction to replace the “it”.

c. Page 1, line 16: Replace “tight links” with more academic language such as “significant associations”.

Page 1, Line 27: We have used “significant associations” replace the “tight links”.

d. Page 1, line 16: Please remove the “the” before psychological issues.

Page 1, Line 27: We have removed “the” before mental health concerns

e. Page 1, line 17: Consider using “mental health concerns” instead of “psychological issues”. Page 1, Line 27: We have used “mental health concerns” instead of “psychological issues”

f. Page 1, line 18: First mention social media addiction, then follow with the abbreviation (SMA). Page 1, Line 26: We have written “social media addiction (SMA)” when SMA first appear.

g. Page 1, line 19: Please clarify what “these factors” refer to.

Page 1, Lines 31-32�We have clarified these factors (anxiety, depression, self-esteem, FoMO and loneliness) the section of abstract.

h. Page 1: Consider explaining why the combination of anxiety, depression, loneliness, self-esteem, and FOMO was investigated. This should also be clarified in the rationale and discussion sections. Page 1, Lines 26-29: We wish to investigate the reasons that contribute to social media addiction. After examining the literature, we found that these factors highly occurred, so we decided to choose these factors to do a comprehensive analysis. We also briefly explain the reasons on Page 1, Lines 26-29.

i. Page 2, lines 9 – 13: Please rephrase these sentences for clarity, as the arguments currently do not make sense. Language editing may help here. Page 2, Lines 54-58: We have rephrased these sentences to make the text more clearly and meaningful.

Introduction:

1. The introduction/rationale section was approached with interest. Thank you.

2. The rationale section presents valid arguments and the importance of the study is clearly indicated and well-argued.

3. Please ensure that important arguments are supported by citations. Please see:

Reply: We sincerely appreciate the valuable comments. We have checked the literature carefully and added more reference into the revised manuscript. In order to save your effort, we have filled the table with the references you proposed that need to be added to facilitate your review. The serial number is the order of the references in the article. In addition to the supplementary literature in the table, we also supplemented the literature on low self-esteem and social media addiction (25) in Page 4, Line 108.

Table2: Supplementary notes to references

a. Page 2, line 22. Page 3, Line 67: (2)

b. Page 2, line 28. Page 3, Line 73: (4)

c. Page 2, line 29. Page 3, Line 75: (5)

d. Page 3, line 7. Page3 Lines 79-83: We have supplemented the literature on the development of SMA as an important academic issue (11) and introduced specific data in recent years to support it. (10)

e. Page 4, line 4. Page 5, Line 111: (27)

f. Page 17, line 27 Page 24, Line 342: (77)

g. Page 18, line 6. Page 25, Lines 364, 366: (80) (81)

4. Page 3: An argument is made that the number of social media addicts is increasing. What studies or statistics support this argument? The authors have not provided any statistics or references to substantiate this important argument.

Reply: We are appreciative of the reviewers' insightful comments, which has strengthened the article's validity. We completely agree with this recommends, and to support this essential point, and added the statistic to verify the increasing of SMA addicts in Page 3, Lines 80-81.

5. Page 3, line 7: Please avoid the phrase “hot topic”, as it lacks academic tone. Instead, refer to it as “a subject of significant interest”.

Reply: We have revised the “hot topic” to “a subject of significant interest” in Page3 Line 82.

6. Page 3, line 12: The phrase “are fewer opportunities” should perhaps be “there were fewer opportunities,” as the lockdown is now over

Reply: We have revised the “are fewer opportunities” to “there were fewer opportunities,” in Page 3, Line 88.

However, due to the irrelevance of the novel coronavirus to this paper, this supplementary content has been deleted.

7. Is social media addiction a global concern regardless of COVID-19? The rationale has a strong focus on the pandemic period, but it would be beneficial to highlight whether SMA is a global concern outside of the COVID-19 context. It is unclear whether the authors intend to focus solely on the COVID-19 period.

Reply: We would like to extend our sincere gratitude for your thoughtful consideration of our manuscript and the constructive feedback provided. Our study was a meta-analysis conducted outside the context of the pandemic, and the data in all the articles collected were not limited to the COVID-19 period. We agree that mentioning the COVID-19 influence in the introduction was not pertinent, and thus, we have removed all such references to ensure clarity and relevance (in Pages 3-4, Lines 84-94; Page 5, Lines 116-118, 123-126). We greatly appreciate your expert review and the invaluable suggestions you have offered. Your feedback has been crucial in refining our work.

8. Page 4, lines 17 to 18: Please rephrase for better clarity.

Reply: Thank for your precious comment, we have rephrased the sentence to “Since the COVID-19 pandemic has been raging, people spend a lot of time on social media for information focus on updates on the pandemic, prevention methods and effective treatment.”

Upon careful consideration, it has been determined that the content related to the novel coronavirus is not directly pertinent to the scope and focus of this manuscript. Consequently, the supplementary material addressing this topic has been deemed inessential and has been removed to maintain the relevance and coherence of the paper.

9. Page 4, line 22: Why is the emphasis only on developing effective interventions for teenagers? Many studies indicate that university/college students are the highest users of social media. Why not also advocate for effective interventions for these students?

Reply: The selection of the term “teenagers” was our negligence. What we meant was the student group. Moreover, the intervention measures are beyond the research scope of this article. More experiments should be carried out to explore the correct intervention methods. Therefore, we are decided to delete this paragraph in Page 5, Lines 128-129,131.

10. Page 3, line 20: Consider using “mental health issues” rather than “mental issues”.

Reply: Thank you for pointing out the shortcomings in the use of language in the article. We have revised “mental issues” into “mental health issues” in Page 4, Lines 96-97.

11. The rationale for the combination of anxiety, depression, loneliness, self-esteem, and FOMO remains unclear after reading this section.

Reply: The reason for combining these factors in this paper is that it is found that these symptoms are relatively common in patients with social media addiction when consulting data. Moreover, according to other experiments conducted by our research group, these symptoms are also relatively common in clinical practice in patients with Internet addiction. We will summarize the specific reasons in the DISCUSSION part in Page 23, Lines 323-330)

12. Strengthen the rationale section by incorporating more literature to support the arguments presented.

Reply: Thank for your professional and detailed suggestions, we have learned that our article needs more relevant literature support, so we have supplemented the corresponding references in the position you pointed (can be checked in Table 2).

Methodology:

1. The methodology section was discussed properly.

2. Page 5, line 10: Clarify what the six elements refer to. Please elaborate.

Reply: We have clarified what the factors (anxiety, depression, loneliness, self-esteem, and FOMO) refer to in Page 6, Lines 145-146.

3. Page 5, line 24 and line 27: Reference is made to self-esteem in line 24 and then again in line 27. Should self-esteem be replaced with FOMO in line 27?

Reply: We are grateful for your careful review of this article. This error is due to our carelessness, FoMO can’t replace self-esteem. We have made revisions in Page 7, Line 166.

4. The manuscript indicates that 32 studies were included in the meta-analysis. However, only 31 studies are listed in Table 1.

Reply: Thank you for your careful review. We have revised table 1 and table 2 (in Pages 9-13 and Pages 14-16). To ensure that the documents included in the meta-analysis are correctly filled in the form, and to ensure the consistency of the full text.

5. Discrepancies exist between Tables 1 and 2. In Table 1, a study is listed as Fabris (2022) – Italian, which does not appear in Table 2. In Table 2, two studies are listed as Kitiş (2022) – Turkey, and Koc (2013) – Turkey, which do not appear in Table 1. Please ensure that the correct studies are reported on.

Reply: Thank you very much for your careful review of this manuscript. We reviewed the manuscript carefully, then revised Table 1 and Table 2. We made the order of documents in Table 1 and Table 2 (in Pages 9-13 and Pages 14-16) consistent and ensured the consistency of the full text.

Results:

Reply: We sincerely thank your valuable feedback that we have used to improve the quality of our manuscript. For the convenience of reference, the reviewer's comments are listed in the table below and numbered. Our changes to the manuscript are given in blue text. Your comments have improved the quality of the article, and we thank you again for your review and the effort you put into it.

Table 3: A description of modifications to the results section

1. The presentation of the results was done in a systematic and structured manner. The results were presented using figures and tables and were discussed properly.

2. Page 12, line 18: Rather refer to the fact that 1315 studies were excluded rather than “should not have been included”. In Page 17, Line 212: We have used the “have been deleted” to replace the “should not have been included”.

3. Page 13, line 1: Rather refer to “screening” the titles and abstracts of the sources rather than “browsing”. In Page 18, line 214: We have used the “screening” to replace the “browsing”.

4. Page 13, line 2: What does “researches” mean” In Page 18, line 215: We have modified the “researches” to “articles”.

5. Page 13, line 5: Reference is made to 27 studies whereas 32 studies are referred to in the manuscript. In Page 18, line 218: We have changed 27 to the correct number (32 studies).

6. Page 13, line 13: … nine studies that reported…. In Page 19, line 227-232: In the results section, each subsection will provide a detailed account of the number of articles included in the meta-analysis for the associations between SMA and anxiety, depression, loneliness, and FoMO. We have deleted this

paragraphs to avoid redundancy.

7. The rationale for combining anxiety, depression, loneliness, self-esteem, and FOMO remains unclear in this section as well. In Pages 23, Lines 323-330�The reason for combining these factors in this paper is that it is found that these symptoms are relatively common in patients with social media addiction when consulting data. Moreover, according to other experiments conducted by our research group, these symptoms are also relatively common in clinical practice in patients with Internet addiction. We will summarize the specific reasons in the DISCUSSION part (in Pages 23, Lines 323-330)

8. Page 16, line 13: Is the abbreviation SMA missing from the In Page 22, line 300� SMA has been added to the title of the figure.

Discussion:

Reply: Thank you for your helpful feedback on our article. We've made changes based on your suggestions and put them in a table for you to see easily. Your comments are important to us, and we've taken them all into account. Here's the table with your comments on the left and our changes on the right. We really appreciate your ideas to make our article better.

Thanks again for your suggestions

Table 3: A description of modifications to the discussion section

1.The discussion of the findings was done well.

2. Page 16, line 22: Refer back to the earlier comment regarding the use of the term “hot topic”. In Page 23, line 318: We have modified "hot topic" to "a subject of significant interest" according to your previous modification suggestions.

3. Page 17, line 1: Use “results” instead of “result”. In Page 24, line 332: We have used “results” instead of “result”.

4. Page 17, line 1: Clarify what the authors consider to be weak correlations. For instance, a correlation of .41 could be considered moderate. In Pages 24, Lines 332: We have changed the “weak correlation” to “weak to moderate correlation”

5. Page 17, line 1: What does MPA refer to? In Page 24, line 333: We apologize for the error in our document where "SMA" was mistakenly written as "MPA." The mistake has been corrected

6. Page 17, lines 9 and 10: Should FOMO not form part of this argument? In Page 24, line 341: We agree with your suggestion to incorporate FoMO into this argument.

7. Page 17, line 20: Provide brief information on the countries where the studies were conducted, as well as the sample sizes, similar to how it was done in other sections. In Pages 24-25, line 352-356: We have provided a concise overview of the country in which the study was conducted, along with the sample size in the part of SMA and anxiety and depression.

8. Page 19, line 4: Please rephrase. Page 26, line 379-380: In order to maintain the focus on the strength of the relationship between FoMO and the variables of interest, without drawing comparisons with other factors, we have delected to

---

## [Decision Letter · Decision Letter 1]

8 Jan 2025

Dear Dr. Zhu,

Thank you for submitting your manuscript to PLOS One. After careful consideration, we feel that it has satisfied our scientific requirements for publication.

However, our editorial team have significant concerns about the grammar, usage, and overall readability of the manuscript. PLOS One requires that published manuscripts use language which is 'clear, correct, and unambiguous', see our criteria for publication at https://journals.plos.org/plosone/s/criteria-for-publication#loc-5. We therefore request that you revise the text to fix the grammatical errors and improve the overall readability of the text.

We suggest you have a fluent English-language speaker thoroughly copyedit your manuscript for language usage, spelling, and grammar. If you do not know anyone who can do this, you may wish to consider employing a professional scientific editing service.

Whilst you may use any professional scientific editing service of your choice, PLOS has partnered with both American Journal Experts (AJE) and Editage to provide discounted services to PLOS authors. Both organizations have experience helping authors meet PLOS guidelines and can provide language editing, translation, manuscript formatting, and figure formatting to ensure your manuscript meets our submission guidelines. To take advantage of our partnership with AJE, visit the AJE website (https://www.aje.com/go/plos/) for a 15% discount off AJE services. To take advantage of our partnership with Editage, visit the Editage website (www.editage.com) and enter referral code PLOSEDIT for a 15% discount off Editage services. If the PLOS editorial team finds any language issues in text that either AJE or Editage has edited, the service provider will re-edit the text for free.

Please note that we will not be able to proceed with publication of your manuscript until the concerns above are addressed.

* A copy of your manuscript showing your changes by either highlighting them or using track changes (uploaded as a supporting information file)

* A clean copy of the edited manuscript (uploaded as the new manuscript file)

We look forward to receiving your revised manuscript.

We look forward to receiving your revised manuscript.

Kind regards,

Marianne Clemence, Staff Editor, on behalf of

Tailson Mariano

Academic Editor

PLOS ONE

Journal Requirements:

Reviewers' comments:

Reviewer's Responses to Questions

**Comments to the Author**

Reviewer #1: All comments have been addressed

Reviewer #2: All comments have been addressed

2. Is the manuscript technically sound, and do the data support the conclusions?

Reviewer #1: Yes

Reviewer #2: Yes

3. Has the statistical analysis been performed appropriately and rigorously?

Reviewer #1: Yes

Reviewer #2: Yes

4. Have the authors made all data underlying the findings in their manuscript fully available?

Reviewer #1: Yes

Reviewer #2: Yes

5. Is the manuscript presented in an intelligible fashion and written in standard English?

Reviewer #1: Yes

Reviewer #2: Yes

Reviewer #1: Comments to the Author(s):

Reviewer Comments on Manuscript ID: PONE-24-20428R1 (research article)

Thank you for the opportunity to review the resubmission of this manuscript. I am satisfied with the changes made by the authors, but I identified some minor errors that they could correct.

• Page 5, line 113: I think the word “and” is missing in the sentence. “…have been examined more and were chosen….”

• Page 5, line 123: Remove the word “used” as the word “using” already appears in the sentence.

• Page 5, line 127: I think RSES is omitted from the sentence. Should the sentence read “…were restricted to RSES or SISE….”

• Page 5, line 129: Replace the full stop with a semicolon for consistency.

• Page 5, line 130: Replace the full stop with a semicolon for consistency.

• Page 15, line 180: Replace showed with shows.

• Page 20, line 283: Did you collect 8839 student participants from seven countries?

Reviewer #2: Dear Author/s, thank you for your detailed attention to concerns and suggestions raised to improve the manuscript. I feel the manuscript is substantially improved according to raised concerns and suggestions. please consider minor comments listed below

1. line 197: please carefully revise " state mindfullness" to SMA

2. line 204: "significative" to significant

3. in figure 2,3,4,5 caption what is abbreviated as PUSM? I think this need revision

4. I suggest the author/s go through the document and edit for some typos, grammatical error, and coherence

**Do you want your identity to be public for this peer review?** For information about this choice, including consent withdrawal, please see our Privacy Policy

Reviewer #1: No

Reviewer #2: No

---

## [Author Response · Author response to Decision Letter 2]

11 Feb 2025

To reviewer #1:

Dear reviewer

Thank you very much for your positive feedback. We have carefully revised the manuscript based on your comments. We hope that these revisions and improvements will satisfactorily address the professional issues you raised. Our point-to-point responses are as follows.

• Page 5, line 113: I think the word “and” is missing in the sentence. “…have been examined more and were chosen….”

Response�感谢你专业的意见我们已经在句子中加入and on line 124, page5

• Page 5, line 123: Remove the word “used” as the word “using” already appears in the sentence.

Response We are grateful for your careful review. The word "used" has been removed on line 135, page 6.

• Page 5, line 127: I think RSES is omitted from the sentence. Should the sentence read “…were restricted to RSES or SISE….”

Response We apologize for our oversight. We have added RSES to the sentence on line 140, page 6, to ensure the smoothness and coherence of the article.

• Page 5, line 129: Replace the full stop with a semicolon for consistency.

Response We have replaced the period with a semicolon on line 142, page 6.

• Page 5, line 130: Rep 6.ace the full stop with a semicolon for consistency.

Response We have replaced the period with a semicolon on line 144, page 6.

• Page 15, line 180: Replace showed with shows.

Response�we have replaced showed with shows on line 198, page 16. Thank you again for your time on this manuscript.

• Page 20, line 283: Did you collect 8839 student participants from seven countries?

Response We are deeply grateful for your professional and meticulous feedback. We regret any confusion that may have arisen from our initial phrasing. It was not our intention to imply that we collected the data ourselves; instead, our focus was on analyzing the data presented in the articles. The sentence has been revised to:

"In our meta-analysis, we analysed the correlation between anxiety and SMA. We collected 11 articles, which included a total sample size of 8,839 students from seven countries: China, Poland, Bangladesh, Turkey, Greece, Spain, and Portugal."

Thank you for your meticulous review. We apologize for our oversight. In response to your comments, we have made the necessary corrections to ensure consistency and harmony throughout the manuscript.

Yours sincerely,

Zhang jing

To reviewer #2:

Dear reviewer

We feel great thanks for your professional review work on our article. As you are concerned, there are several problems that need to bead dressed. According to your nice suggestions. we have made corrections to our manuscript and the detailed corrections are listed below.

1. line 197: please carefully revise " state mindfullness" to SMA

Response Thanks for your careful review, we have revised it to on line 218, page 17.

2. line 204: "significative" to significant

Response Thank you for your suggestion. We have revised "significative" to "significant" on line 226, page 18.

3. in figure 2,3,4,5 caption what is abbreviated as PUSM? I think this need revision

Response We are thankful for your careful review. We have modified Figure 2,3,4,5 and re-uploaded it to the system.

4. I suggest the author/s go through the document and edit for some typos, grammatical error, and coherence

Response Thank you for your professional review. We apologize for our careless. In response to your comments, we have made the necessary corrections to ensure consistency and harmony throughout the manuscript. Consequently, we have decided to engage the services of the professional editing website AJE to further polish our article, with the hope that it will fully meet your requirements.

Thank you very much for your attention and time. Look forward to hearing from you.

Yours sincerely,

Zhang jing

---

## [Decision Letter · Decision Letter 2]

22 Apr 2025

Dear Dr. Zhu,

Thank you for submitting your manuscript to PLOS ONE. After careful consideration, we feel that it has merit but does not fully meet PLOS ONE’s publication criteria as it currently stands. Therefore, we invite you to submit a revised version of the manuscript that addresses the points raised during the review process.

We look forward to receiving your revised manuscript.

Kind regards,

Tailson Mariano, Ph.D.

Academic Editor

PLOS ONE

Journal Requirements:

Reviewers' comments:

Reviewer's Responses to Questions

**Comments to the Author**

Reviewer #3: All comments have been addressed

Reviewer #4: All comments have been addressed

2. Is the manuscript technically sound, and do the data support the conclusions?

Reviewer #3: Yes

Reviewer #4: Yes

3. Has the statistical analysis been performed appropriately and rigorously?

Reviewer #3: Yes

Reviewer #4: Yes

4. Have the authors made all data underlying the findings in their manuscript fully available?

Reviewer #3: Yes

Reviewer #4: Yes

5. Is the manuscript presented in an intelligible fashion and written in standard English?

Reviewer #3: Yes

Reviewer #4: Yes

Reviewer #3: (No Response)

Reviewer #4: The article has scientific and mainly social relevance, considering the phenomenon of problematic behavior related to social networks. It is a topical issue that needs attention due to its negative effects on individuals. Understanding the variables involved in this construct is important for us researchers; however, in the text, these variables need to be better stitched together. Below are some suggestions for improving the manuscript.

Introduction:

Line 58: Introduce global data and then specify China.

Lines 62 and 64: The phrase "On the one hand" is repeated.

Line 79: The sentences dealing with depression and self-esteem need to be better connected.

Line 86: A linking paragraph is needed to talk about loneliness. The transition is abrupt.

Line 99: Provide social, theoretical, and methodological justification for the study. The variables of interest need to be better connected. The objectives need to be clearly presented.

Discussion:

Line 249: Start by restating the objectives of the study.

Limitations:

Present the limitations and also the possible solutions, and how they can be remedied.

**Do you want your identity to be public for this peer review?** For information about this choice, including consent withdrawal, please see our Privacy Policy

Reviewer #3: No

Reviewer #4: No

---

## [Author Response · Author response to Decision Letter 3]

26 May 2025

Dear Reviewers:

We sincerely appreciate your valuable comments. In response to your suggestions, we have carefully reviewed the manuscript and accordingly implemented the following modifications:

(1) Line 58: Introduce global data and then specify China.

We thank you for this constructive suggestion. To enhance the logical flow of the paper, we have accordingly supplemented the manuscript with the following additions, accompanied by proper citation to the data sources:

As of January 15, 2025, there were 5.40 billion internet users, equivalent to 66% of the world’s population. And the China Internet Network Information Center (CNNIC) published the 51st Statistical Report on the Development Status of the Internet in China in March 2022. (in page 2, lines 58-61)

(2) Lines 62 and 64: The phrase "On the one hand" is repeated.

We appreciate your insightful feedback. To improve textual coherence, we have rephrased the indicated passage as follows:

On the one hand, due to the large influx of information, internet users are constantly confirming the validity and trustworthiness of information on social media; on the other hand, this feature increases dependence on social media owing to the increased demand for information. (in pages 2-3, lines 63-67)

(3) Line 79: The sentences dealing with depression and self-esteem need to be better connected.

We are particularly grateful for this crucial observation. In response to your comment, we have substantially strengthened the conceptual linkage between depression and self-esteem in the revised version, with the following key enhancements:

Self-esteem is a subjective evaluation that refers to how people feel about themselves and numerous studies have shown a negative correlation between SMA and self-esteem, which may be related to the greater need for people with low self-esteem to cultivate identity through social media. (in page 3, lines 82-85)

(4) Line 86: A linking paragraph is needed to talk about loneliness. The transition is abrupt.

This perspicacious critique has brought to light critical structural inconsistencies in our original exposition. With profound gratitude for your scholarly vigilance, we have implemented revisions that fundamentally reconstitute the argumentation framework through:

Meanwhile, with the advancement of society and the deepening division of labor, loneliness has emerged as a significant contributing factor to SMA. After extensive research, researchers classified loneliness according to its causes, which can be divided into emotional loneliness and social loneliness. (in page 4, lines 89-91)

(5) Line 99: Provide social, theoretical, and methodological justification for the study. The variables of interest need to be better connected. The objectives need to be clearly presented.

We are profoundly grateful for your seminal critique regarding the conceptual scaffolding of our literature review. In direct response to your astute observations, we have conducted a refinement of the research motivation:

According to the published research, anxiety, depression, loneliness, FoMO and self-esteem are the most common factors in SMA among adolescents. This article aims to explore the relationship between SMA and anxiety, depression, loneliness, FoMO and self-esteem, in order to understand the etiology of SMA and provide new approaches for prevention or treatment. As a populous country and a developing country that is rapidly digitizing, China is confronted with a severe challenge: the sharp increase in the number of teenagers suffering from SMA has escalated into an urgent public health issue. Therefore, studying the psychological roots of this phenomenon has become an urgent priority. Based on the above purposes, this study can deepen our understanding of addictive behaviors and help prevent adverse effects on students' physical and mental health. (in pages 4-5, lines 105-115)

(6) Start by restating the objectives of the study.

We really appreciate the effort you have put into this article. In response to your professional and detailed comments, we have made improvements: Furthermore, this article can provide more comprehensive guidance for various approaches to intervention, government policy-making and the classification of addictive behavior. Enhance the public's awareness of preventing Internet addiction, improve their online media literacy and protection skills, and ensure healthy and civilized Internet use. (in page 21, lines 294-298)

(7) Present the limitations and also the possible solutions, and how they can be remedied.

We sincerely appreciate your insightful comments, which have helped us better recognize the limitations of our current study. Through careful reflection, we have identified several key areas requiring improvement in future research:

However, methodological constraints include (a) underpowered sample cohorts in the meta-analysis, and (b) ethnocentric recruitment practices in the source studies, failing to represent cross-cultural populations. We will do our best to make improvements in the future work. (in page 26, lines 403-406)

Once again, we would like to express our gratitude to the reviewers for their insightful comments and to you for overseeing the review process. We hope that our revisions meet with your approval and that our manuscript can be considered for publication in your esteemed journal.

Best regards,

Zhangjing

---

## [Decision Letter · Decision Letter 3]

17 Jul 2025

Correlations between social media addiction and anxiety, depression, FOMO, loneliness and self-esteem among students: A systematic review and meta-analysis

PONE-D-24-20428R3

Dear Dr. Zhu,

We’re pleased to inform you that your manuscript has been judged scientifically suitable for publication and will be formally accepted for publication once it meets all outstanding technical requirements.

Kind regards,

Tailson Evangelista Mariano, Ph.D.

Academic Editor

PLOS ONE

Additional Editor Comments (optional):

Reviewers' comments:

Reviewer's Responses to Questions

**Comments to the Author**

Reviewer #4: All comments have been addressed

2. Is the manuscript technically sound, and do the data support the conclusions?

Reviewer #4: Yes

3. Has the statistical analysis been performed appropriately and rigorously?

Reviewer #4: Yes

4. Have the authors made all data underlying the findings in their manuscript fully available?

Reviewer #4: Yes

5. Is the manuscript presented in an intelligible fashion and written in standard English?

Reviewer #4: Yes

Reviewer #4: Dear authors,

First and foremost, I would like to commend you for your dedication and effort in improving the manuscript. I am pleased with the revisions made, as it is evident that you have thoughtfully considered my suggestions. I would like to reaffirm the significance of this article for both the scientific community and society as a whole. I encourage you to continue investigating these important topics that have a profound impact on modern society.

**Do you want your identity to be public for this peer review?** For information about this choice, including consent withdrawal, please see our Privacy Policy

Reviewer #4: No

---

## [Editor Report · Acceptance letter]

PONE-D-24-20428R3

PLOS ONE

Dear Dr. Zhu,

I'm pleased to inform you that your manuscript has been deemed suitable for publication in PLOS ONE. Congratulations! Your manuscript is now being handed over to our production team.

Kind regards,

on behalf of

Dr. Tailson Evangelista Mariano

Academic Editor

PLOS ONE